# BMSCs differentiated into neurons, astrocytes and oligodendrocytes alleviated the inflammation and demyelination of EAE mice models

**Guo-yi Liu[1], Yan Wu[1], Fan-yi Kong[2], Shu Ma[2], Li-yan Fu[2], Jia Geng[1] ***

**1** Department of Neurology, The First Affiliated Hospital of Kunming Medical University, Kunming, Yunnan Province, P R China, **2** Department of Neurology, 920th Hospital of Logistics Support Force, People's Liberation Army. No. 212, Kunming, Yunnan Province, P R China

\* basler777@163.com

**Data Availability Statement:** All relevant data are within the paper.

**Funding:** This work was supported by the National Natural Science Foundations of China (81760226),

## Abstract

Multiple sclerosis (MS) is a complex, progressive neuroinflammatory disease associated with autoimmunity. Currently, effective therapeutic strategy was poorly found in MS. Experimental autoimmune encephalomyelitis (EAE) is widely used to study the pathogenesis of MS. Cumulative research have shown that bone marrow mesenchymal stem Cells (BMSCs) transplantation could treat EAE animal models, but the mechanism was divergent. Here, we systematically evaluated whether BMSCs can differentiate into neurons, astrocytes and oligodendrocytes to alleviate the symptoms of EAE mice. We used Immunofluorescence staining to detect MAP-2, GFAP, and MBP to evaluate whether BMSCs can differentiate into neurons, astrocytes and oligodendrocytes. The effect of BMSCs transplantation on inflammatory infiltration and demyelination in EAE mice were detected by Hematoxylin-Eosin (H&E) and Luxol Fast Blue (LFB) staining, respectively. Inflammatory factors expression was detected by ELISA and RT-qPCR, respectively. Our results showed that BMSCs could be induced to differentiate into neuron cells, astrocytes and oligodendrocyte *in vivo* and *in vitro*, and BMSCs transplanted in EAE mice were easier to differentiate than normal mice. Moreover, transplanted BMSCs reduced neurological function scores and disease incidence of EAE mice. BMSCs transplantation alleviated the inflammation and demyelination of EAE mice. Finally, we found that BMSCs transplantation down-regulated the levels of pro-inflammatory factors TNF-α, IL-1β and IFN-γ, and up-regulated the levels of anti-inflammatory factors IL-10 and TGF-β. In conclusion, this study found that BMSCs could alleviate the inflammatory response and demyelination in EAE mice, which may be achieved by the differentiation of BMSCs into neurons, astrocytes and oligodendrocytes in EAE mice.

## Introduction

Multiple sclerosis (MS) is a chronic neuroinflammatory disease that is associated with autoimmunity in central nervous system (CNS) [1,2]. Generally speaking, it is characterized by axon

Yunnan health training project of high level talents (D-2018029) and Yunnan Applied Basic Research Projects [2018FE001(-145)].

**Competing interests:** The authors have declared that no competing interests exist.

**Abbreviations:** BMSCs, Bone marrow mesenchymal stem Cells; CNS, Central nervous system; EAE, Experimental autoimmune encephalomyelitis; GFAP, Gial fibrillary acidic protein; LFB, Luxol Fast Blue; MAP-2, Microtubule-associated protein 2; MBP, Myelin Basic Protein; MS, Multiple sclerosis; O4, Oligodendrocyte Marker.

damage, demyelination, inflammatory infiltration and progressive neurological damage, eventually leading to disability [3]. Moreover, MS is thought to be a multifocal demyelination disease in CNS [4]. Currently, MS mainly depends on three types of drug treatment: disease-modifying drugs (DMD) specifically designed for MS, corticosteroids for acute exacerbations, and drugs for symptomatic control. However, it is less evident that drugs are effective in the progression of MS [5]. Thus, to find an effective therapeutic strategy is very important for MS in clinical. Experimental autoimmune encephalomyelitis (EAE) is widely used to study the pathogenesis of MS, which represents both pathological and features of MS [6]. To some extent, EAE could effectively elucidate various pathological processes in MS, such as inflammation demyelination, axonal lesions [7]. Bone marrow mesenchymal stem Cells (BMSCs) are non-hematopoietic stromal cells that derived from bone marrow [8]. BMSCs differentiate into various cell types, which contribute to regeneration of tissues [9–11]. BMSCs also play immunomodulatory role through inhibiting T-cell activities [12]. In addition, BMSCs secrete growth factors to regulate hematopoietic stem/progenitor cell proliferation and differentiation [13]. Recently, some researchers have shifted the focus to stem cell-based therapy in many diseases, including nervous system disease, indicating that stem cell may be potentially amenable to therapeutic manipulation for clinical application of MS [14,15]. In the acute and subacute phases, the tissues were selectively target damaged by intravenous injection of BMSCs, which improved the recovery of neurological function, decreased inflammatory response and demyelination after EAE [16]. However, whether BMSCs have a therapeutic effect on EAE mice through differentiation remains to be studied. In addition, it is now recognized that the interaction of neurons, astrocytes and oligodendrocytes plays an important regulatory role in remyelination and the development of and MS [17,18]. Therefore, we explored whether the transplanted BMSCs can differentiate into neurons, astrocytes and oligodendrocytes to alleviate the development of MS. In our research, we found that BMSCs could differentiate into neurons, astrocytes and oligodendrocytes *in vivo* and *in vitro*, and transplanted BMSCs could alleviate the inflammatory response and demyelination of EAE mice, and improve clinical symptoms.

## Materials and methods

### Animal

Male C57BL/6 mice aged 8–10 weeks were purchased from Kunming Medical University. The mice were housed in sterile, constant temperature rooms at Kunming Medical University with a 12 h/12 h light/dark cycle, with free access to food and water. All animal experiments were conducted according to the ARRIVE guidelines, and were performed in accordance with Ethics Committee of Kunming Medical University and approved by Ethics Committee of Kunming Medical University. All measures had been taken to minimize the suffering of animals. Compared to the starting point of EAE immunization, no animal lost more than 20% of its body weight and no neurological score exceeded 4. The health of the mice was monitored at least once a day, and no accidental deaths were observed. Euthanasia was carried out in a $CO_2$ chamber, gradually filled with $CO_2$ before 4 of mice neurological score, and then bled. All animal facility staff and researchers had followed required courses and obtained certificates to conduct animal research.

### Isolation and identification of primary BMSCs

According to the method of Gao et al. [19], BMSCs were isolated from male mice anesthetized (1% sodium pentobarbital, 40 mg/kg intraperitoneal injection). After being disinfected with 75% alcohol, the femur and tibia were obtained by removing the surface muscles and fascia.

Bone marrow was then obtained by rinsing with Hanks solution. After 5 min of centrifugation at 170 g, BMSCs at the bottom of the centrifuge tube were remixed evenly with Hanks solution. Followed by 5 min of centrifugation at 1050 g to obtained BMSCs. Isolated BMSCs were cultured at 37˚C and 5% $CO_2$ with DMEM/F12 medium (Gibco, USA) containing 10% fetal bovine serum (FBS; Gibco, USA), 100 U/ml penicillin and 100 U/ml streptomycin. In our previous study, we have evaluated phenotype of BMSCs by flow cytometric analysis. To directly track BMSC differentiation *in vivo*, the BMSCs were labeled with a lentiviral vector encoding enhanced GFP (Cyagen, USA). Briefly, BMSCs ($5\times103$ cells/cm$^2$) were incubated in 6-well plates for 24 h. Then the culture medium was removed and concentrated viral supernatant diluted in serum-free α-MEM was added. Eight hours later, the viral supernatant was replaced with complete culture medium. G418 (100 μg/ml) was used to purify the GFP-positive cells.

## EAE model

Mice were randomly divided into NC, NC+BMSCs, EAE and EAE+BMSCs groups, with 15 mice in each group. Mice of EAE and EAE+BMSCs groups performed EAE modeling. In short, Mice were immunized subcutaneously with 200 μg $MOG_{35-55}$ (peptide sequence: Met-Glu-Val-Gly-Trp-Tyr-Arg-Ser-Pro-Phe-Ser-ArgVal-Val-His-Leu-Tyr-Arg-Asn-Gly-Lys; MedChem Express, US) peptide emulsified in complete Freund's adjuvant (Sigma, USA) containing *Mycobacterium tuberculosis* H37Ra (BD Biosciences, USA) on 0 day. Then mice were injected intravenously with 300 ng Pertussis Toxin (Millipore, USA) both immediately after immunization and 2 days later. In 8 days post-EAE induction, BMSCs were injected into the lateral ventricle (Start with the dura mater; before and after the halogen: 0.6 mm; opening: 1.5 mm; depth: 1.7 mm) of mice of NC+BMSCs and EAE+BMSCs groups by brain stereotactic technology.

## Induced BMSCs to differentiate into neurons, astrocytes and oligodendrocytes

BMSCs from the fifth generation were seeded in 6-well plates coated with poly ornithine and laminin at a density of $2\times10^5$ cells/mL. After 24 h, DMEM/F12 medium performs the following replacements. For neuron induction, it was replaced with DMEM/F12 medium contained 20% FBS, 6 mM β-Mercaptoethanol, 20 ng/ml bFGF, 1.7 μM rhSHH, 100 ng/ml FGF-8, 0.3 μg/ml all-trans retinoic acid. After 14 d, it was replaced with differentiation maintenance solution (DMEM/F12 medium contained 1%N2 and 2% B27). For astrocyte induction, it was replaced with NS medium contained 2% B27, 2 mM glutamine, 50 U/ml penicillin, 50 μg/ml streptomycin and 60 μg/ml T3. After 14 d, it was replaced with differentiation maintenance solution. For oligodendrocyte induction, it was replaced with NS medium contained 2% B27, 2 mM glutamine, 50 U/ml penicillin, 50 μg/ml streptomycin and 60 μg/ml T3. The complete culture medium was changed every 2 to 3 days. BMSCs in control group were not induced. After 21 days, the induced neurons, astrocytes and oligodendrocytes were harvested for follow-up experiments.

## Clinical signs measurement

During the *in vivo* experiment, the weight, neurobehavioral score and disease incidence of the mice were monitored by two independent observers in a blind manner until the 36th day after immunization. The neurobehavioral score is defined by the following scale: no clinical symptoms = 0; wobbled gait or loss of tail tension = 1; tail weakness, hind limb weakness = 2; hind limb paralysis = 3; hind limb paralysis, forelimb paralysis or Forelimb weakness with bowel dysfunction = 4; dying or death = 5; the symbol between them is ±0.5 [20]. In our preliminary

**Table 1. Primary antibodies.**

| Primary antibodies | Company | Dilution |
|---|---|---|
| MAP-2 | Abcam | 1:600 |
| βIII-tubulin | Abcam | 1:400 |
| GFAP | Abcam | 1:300 |
| MBP | Abcam | 1:400 |
| O4 | Sigma | 1:500 |

experiments, the peak of the disease is from the 9th to the 15th day after immunization. After 16th days after immunization, the mice were euthanized by cervical dislocation. We collected serum from the mice and took cortex and hippocampus of brain tissues and lumbar spinal cord for the following experiments.

## Immunofluorescence staining (IF)

After fixed with 4% paraformaldehyde, differentiated BMSCs and brain frozen section (10 μm) were penetrated with PBS containing 0.4% Triton X-100. Then, the cells/brain sections were blocked with 5% bovine serum albumin (BSA; Beijing; China) at room temperature for 1 h. The primary antibodies were incubated overnight at 4˚C. The secondary antibodies with fluorophores (1:1000, KPL) were incubated at 4˚C for 1 h. Then, nuclei staining with DAPI followed by capturing using a microscope (Olympus, Japan). We calculated the ratio of BMSCs to differentiate into neurons, astrocytes and oligodendrocytes, respectively. The primary antibodies were shown in Table 1.

## Luxol Fast Blue staining (LFB)

Spinal cord sections of Lumbar spine were placed in a xylene solution for gradient hydration. Then, the sections were put into Luxol fast blue dye solution (Sigma, USA) in a 50–65˚C incubator overnight. Stained sections were taken out and passed through alcohol, water, and then added the color separation solution. Sections were washed 3 times with water after differentiation, and then counterstained or gradient dehydrated., Each mice analyzed 3 histological sections and calculated the average score in our experiment. Standard score for the degree of demyelination: none = 0; rare lesions = 1; demyelination of a few areas = 2; demyelination of large areas or fusion areas = 3 [21].

## Hematoxylin-Eosin staining (H&E)

The mice were perfused with normal saline and 4% paraformaldehyde through the blood vessel. The lumbar spinal cord was fixed with 4% paraformaldehyde for 24 h, embedded in paraffin, and cut into 4–5 μm thick sections. H&E staining was performed according to the manufacturer's instructions to assess the degree of inflammatory cell infiltration. In our experiment, each mice analyzed 3 histological sections and calculated the average score. Standard scores for the degree of inflammatory infiltration: no infiltrating cells = 0; a small amount of scattered infiltrating cells = 1; inflammatory infiltrating tissue around blood vessels = 2; extensive perivascular scar infiltration = 3 [21].

## RNA extraction and Real-time quantitative PCR (RT-PCR)

According to the instructions, total RNA of cells and brain tissues were extracted using Trizol Reagent (Lifetech, USA). We followed the instructions of the FastKing RT Kit (Fermentas;

**Table 2. Primer sequences.**

| Gene | Sequence (5'~3') | Company |
|---|---|---|
| GFAP | AATCACAAGGTCACAAGA | Invitrogen |
| | GGCGTTCCATTTACAATC | |
| βIII tubulin | TCAAGATGTCCTCCACCTT | Invitrogen |
| | GTGAACTGCTCGGAGATG | |
| MAP-2 | AAGGTGAACAAGAGAAAGA | Invitrogen |
| | GAGAAGGAGGCAGATTAG | |
| MBP | ACTATAAATCGGCTCACAAG | Invitrogen |
| | AGCGACTATCTCTTCCTC | |
| O4 | CCTTGTTGCCACCATCTG | Invitrogen |
| | CATACAGGGAGTAGCCAAAG | |
| GAPDH | AAAGGGTCATCATCTCTG | Invitrogen |
| | GCTGTTGTCATACTTCTC | |

Shanghai, China) to synthesize the first strand of cDNA. Then, we performed quantitative PCR by SYBR Green master mix (KAPA; Shanghai, China). The primers were designed using beacon designer 7.90. The primer sequences were listed in Table 2. All experimental results were analyzed by $2^{-\Delta\Delta Ct}$ method.

## ELISA assay

The levels of inflammatory factors TNF-α, IL-1β and IFN-γ, and anti-inflammatory factors IL-10 and TGF-β were detected by ELISA. According to the instructions of ELISA Kit (R&D Systems, USA), the supernatant was collected by centrifugation homogenate (5000rpm for 15 minutes at 4˚C). The levels of TNF-α, IL-1β, IFN-γ, IL-10 and TGF-β were determined by ELISA Kit. We constructed standard curves by standard samples. Quantification of ELISA results were performed at 450 nm using an ELISA plate reader (Spectra Max 190, Molecular Devices, USA).

## Statistical analysis

GraphPad Prism 7 software (GraphPad, USA) was used to conduct statistical analysis. One-way ANOVA and $t$ test were used to analyze data. Data were presented as mean ± standard deviation (SD) and $P$ value$<$0.05 were considered as significant results.

# Results

## Identification of primary BMSCs

Before the BMSCs experiment, we used an inverted phase contrast microscope and flow cytometry to detect the morphology and marker expression levels to identify BMSCs. Observed under an inverted phase contrast microscope, the BMSCs cells of the P1 generation adhered to the wall and presented a polygonal shape. After growing to the P3 generation, the overall BMSCs cells showed a swirl shape at high density, while BMSCs cells gradually showed a fibroblast-like spindle shape at low density (Fig 1A). The above observation results are consistent with the basic morphological characteristics of BMSCs. Furthermore, we detected the levels of BMSCs related markers CD29, CD90, CD34 and CD45 by flow cytometry. The result showed that cultured BMSCs expressed positive CD29$^+$ (94.55±1.67%) and CD90$^+$ (94.67

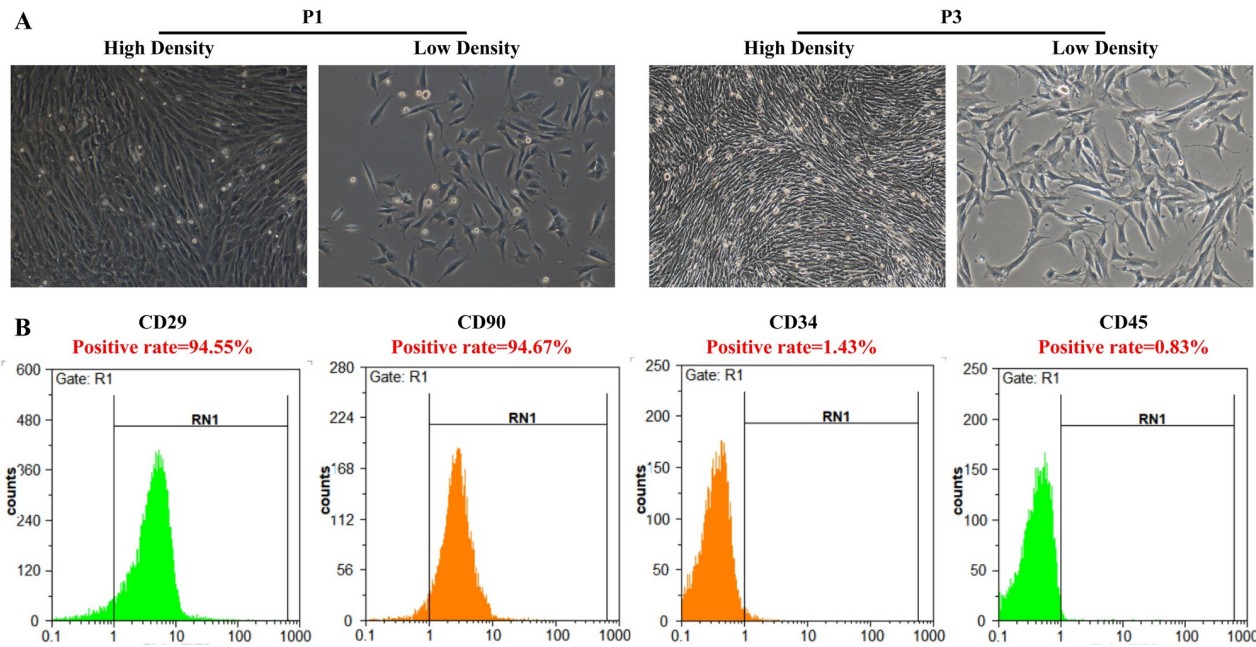

**Fig 1. Identification of primary BMSCs.** (A) High density and low density of P1 and P3 BMSCs morphology was assessed by an inverted phase contrast microscope. Original magnification: 100×. (B) CD29, CD34, CD45 and CD90 surface marker expression of BMSCs was determined by flow cytometry.

±1.88%), and negative CD34⁻ (1.43±0.19%) and CD45⁻ (0.83±0.08%) on the surface of BMSCs (Fig 1B). All the results were consistent with the previous studies [22,23].

## BMSCs were induced to differentiate into neurons, astrocytes and oligodendrocytes

To verify that BMSCs can differentiate into neurons, astrocytes and oligodendrocytes, we induced the culture of BMSCs, and used IF and RT-qPCR to detect the expression level of neurons markers (MAP-2 and βIII-tubulin), astrocytes markers (GFAP) and oligodendrocytes markers (MBP and O4), respectively. The results of IF showed that MAP-2 and βIII-tubulin were expressed in BMSCs cultured in neurons induction medium (Fig 2A). GFAP was expressed in BMSCs cultured in astrocytes induction medium (Fig 2C). MBP and O4 were expressed in BMSCs cultured in oligodendrocytes induction medium (Fig 2E), and the differentiation ratio of BMSCs was above 95%. In addition, the RT-qPCR results showed that compared with the NC group treated with PBS, the expression levels of MAP-2 and βIII-tubulin in the Induced neurons group (Fig 2B), GFAP in the Induced astrocytes group (Fig 2D), and MBP and O4 in the Induced neurons group (Fig 2F) significantly upregulated. It could be seen from the above experimental results that BMSCs could be induced to differentiate into neurons, astrocytes and oligodendrocytes *in vitro*.

## The transplanted BMSCs differentiated into neurons, astrocytes and oligodendrocytes in the EAE mice

To verify that BMSCs can differentiate into neurons, astrocytes and oligodendrocytes *in vivo*, we transplanted GFP-labeled BMSCs into EAE mice model by brain stereotactic technology,

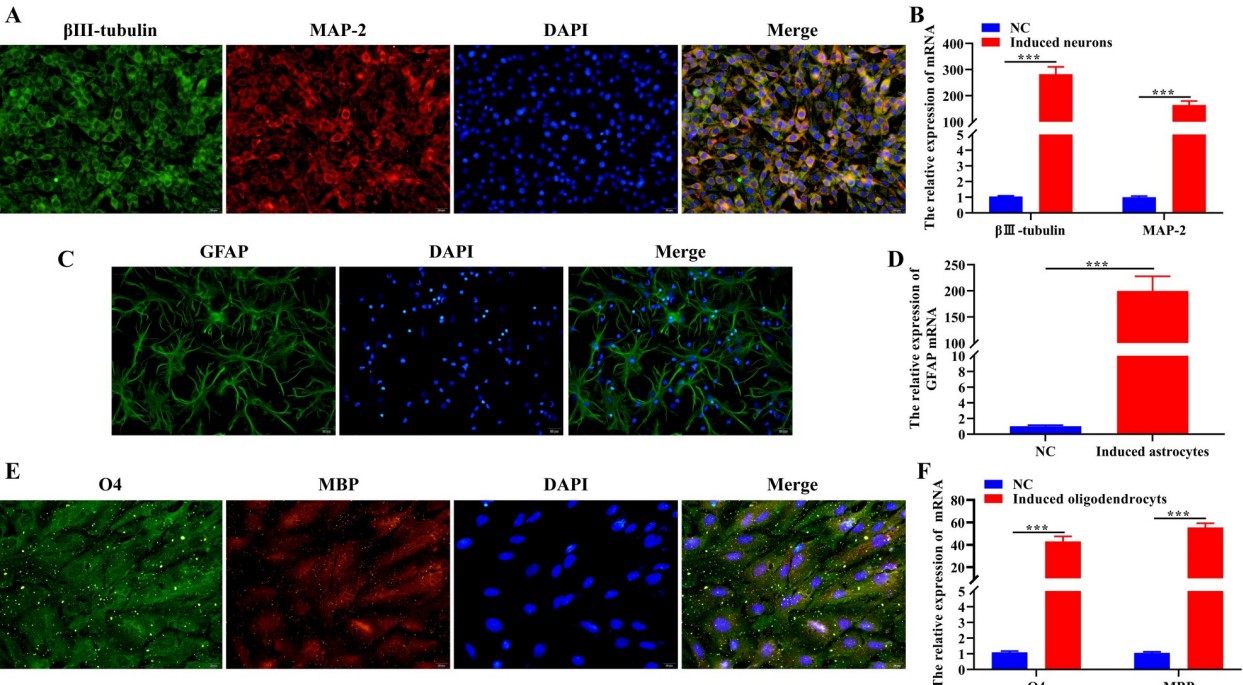

**Fig 2. BMSCs were induced to differentiate into neurons, astrocytes and oligodendrocytes.** Respectively, 21 days after the induction of BMSCs, IF staining was performed to detected the position and level of (A) neurons (MAP-2 and βIII-tubulin), (C)astrocytes (GFAP), and (E) oligodendrocytes marks (MBP and O4). Scale bar = 20 μm. The expression levels of (B) MAP-2 and βIII-tubulin, (D) GFAP, (F) MBP and O4 mRNA were detected by RT-qPCR. In all cases, Values are mean ± SD (n = 3 for each group; $^{**}P<0.01$, $^{***}P<0.001$).

and observed the differentiation of BMSCs in hippocampus and cortex through IF. As shown in Fig 3A, the GFP fluorescence signal could be observed in NC+BMSCs and EAE+BMSCs group, while the NC and EAE group had no GFP fluorescence signal at all. In addition, after merge, it was found that the expression positions of MAP-2, GFAP and MBP overlap with the fluorescent signal positions of GFP in the hippocampus and cortex of NC+BMSCs and EAE +BMSCs groups. It is indicated that transplanted BMSCs could differentiate into neurons, astrocytes and oligodendrocytes in the normal and EAE mice. Furthermore, we conducted a statistical analysis of the IF results and found that the fluorescence intensities of MAP-2, GFAP and MBP in the hippocampus and cortex of EAE+BMSCs group were significantly higher than those in EAE group, and there was no significant difference between NC and NC +BMSCs groups (Fig 3B). Further, we analyzed the differentiation of BMSCs. The results showed that the relatively differentiated BMSCs in the EAE+BMSCs group were significantly higher than NC+BMSCs (Fig 3C). It is suggested that the transplanted BMSCs could increase the number of neurons, astrocytes and oligodendrocytes in the hippocampus and cortex in EAE model. This result was at least partly due to the differentiation of BMSCs into neurons, astrocytes and oligodendrocytes.

## Effect of BMSCs transplantation on clinical signs during EAE progression in mice

To determine whether the differentiation of BMSCs can alleviate the clinical symptoms of EAE, we performed disease incidence analysis, neurobehavioral score and weight measurement on EAE mice and BMSCs transplanted EAE mice. A higher neurobehavioral score

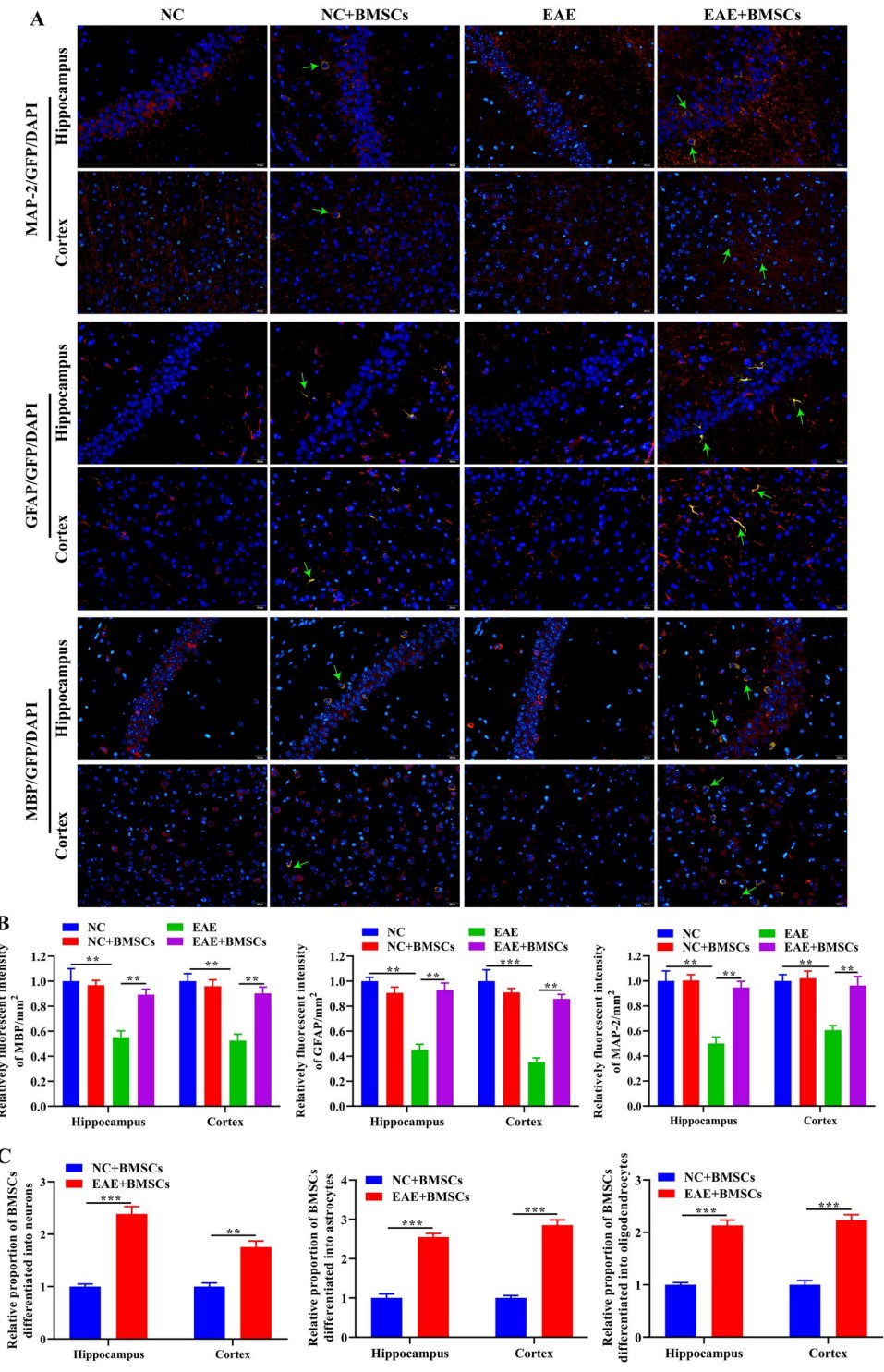

**Fig 3. The transplanted BMSCs differentiated into neurons, astrocytes and oligodendrocytes in the EAE mice.** On the 16th day post-immunization, the mice of each group were euthanized. (A) IF staining exhibited the position and level of GFP with MAP-2, GFAP and MBP in hippocampus and cortex. Scale bar = 20 μm. The green arrows indicated neurons, astrocytes and oligodendrocytes differentiated by BMSCs. (B) The result of MAP-2, GFAP and MBP fluorescence intensity were analyzed by Image J software. (C) In the NC+BMSCs and EAE+BMSCs groups, relative proportion of BMSCs differentiated into neurons, astrocytes and oligodendrocytes, respectively. In all cases, Values are mean ± SD (n = 15 for each group; **P<0.01, ***P<0.001).

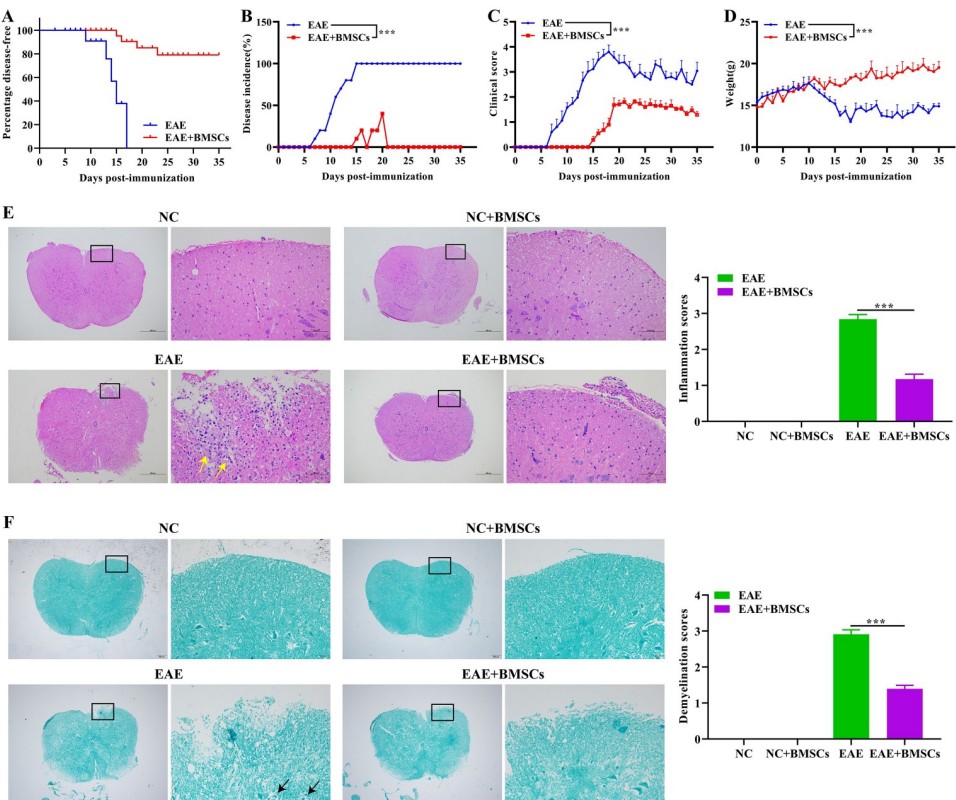

**Fig 4. Effect of BMSCs transplantation on clinical signs and histological changes during EAE progression in mice.**
The body weight and clinical scores of all of mice were assessed daily according to the same criteria for 35 continuous days. (A) Percentage disease-free, (B) disease incidence, (C) clinical scores and (D) body weight, were monitored. On the 16th day post-immunization, (E) H&E staining was used to observe inflammatory infiltration in the lumbar spinal cord of mice, and result of inflammation score. Scale bar = 500 or 100 μm. The yellow arrow indicated inflammatory cell infiltration. (F) LFB staining showed demyelination in the lumbar spinal cord of mice, and result of demyelination score. Scale bar = 500 or 100 μm. The black arrow indicated demyelination. In all cases, Values are mean ± SD (n = 15 for each group; ***P<0.001).

indicates worsening of motor dysfunction. Higher disease incidence and lower body weight indicate an increase in disease severity. The line chart shown that, in 10th-15th post-immunization, mice in the EAE group was large-scale disease, and in 15th post-immunization mice, all mice in the EAE group became sick. In the EAE+BMSCs group, mice begin to develop symptoms from the 14th day post-immunization, and the onset of the mice tended to be stable from the 21st day post-immunization (Fig 4A and 4B). The neurobehavioral score of the EAE group increased rapidly after immunization, and reached a peak on the 18th day, and BMSCs treatment could significantly reduce the neurobehavior score of the EAE in mice (Fig 4C). Furthermore, the mice weight was decreased significantly on the 10th day after immunization in the EAE group. Interestingly, the body weight of mice had been showing an upward trend in the EAE+BMSCs group (Fig 4D). In summary, the differentiation of transplanted BMSCs could alleviate clinical signs during EAE progression in mice.

## Effect of BMSCs transplantation on histological changes during EAE progression in mice

Mice was euthanasia on the 16th day after immunization, collected the lumbar spinal cord tissues of mice for histological analysis to analyze the progression of the disease at the level of

CNS damage. The results of H&E (Fig 4E) and LFB (Fig 4F) shown the inflammatory cell infiltration and demyelination status of mice. The results shown that a lot of inflammatory cells gathered around the small blood vessels in the lumbar spinal cord of the EAE group, and the degree of demyelination of the lumbar spinal cord of the EAE group was significantly higher. However, after BMSCs transplantation, the degree of infiltration of inflammatory cells in the lumbar spinal cord of the EAE+BMSCs group was alleviated, and the degree of demyelination was also significantly reduced. Moreover, In the NC and NC+BMSCs groups, the inflammation and demyelination scores of the mice spinal cord were both 0. It could be seen that BMSCs transplantation could alleviate the infiltration of inflammatory cells and demyelination in the EAE mice.

## Effect of BMSCs transplantation on inflammatory factor expression during EAE progression in mice

In the development of MS, TNF-α, IL-1β and IFN-γ played important roles in infiltration of inflammatory cells and demyelination, while IL-10 and TGF-β could alleviate the progression of MS. Therefore, we detected the expression levels of these inflammatory factors in the serum of mice by ELISA, and further detected their mRNA expression in spinal cord tissue by RT-qPCR. ELISA results shown that in the EAE group, the levels of TNF-α, IL-1β and IFN-γ were significantly up-regulated, whereas IL-10 and TGF-β levels exhibited an opposite pattern. After BMSCs transplantation, the levels of TNF-α, IL-1β, IFN-γ, IL-10 and TGF-β were restored. There was no significant difference in the levels of these inflammatory factors in NC and NC+BMSCs groups (Fig 5A). As expected, RT-qPCR results shown that, compared with NC and NC+BMSCs groups, mRNA expression of TNF-α, IL-1β and IFN-γ in the EAE group were significantly up-regulated, while mRNA expression of IL-10 and TGF-β was significantly decreased. Moreover, the transplanted BMSCs could restore expression of these mRNAs in the spinal cord of mice (Fig 5B). These results were consistent with previous results of alleviating inflammation.

## Discussion

MS is a complex neuroinflammatory disease caused by local inflammation and immune dysfunction, leading to demyelination and extensive mononuclear cell infiltration [24–26]. Generally speaking, MS is considered as a disease of central nervous system involved in CD4[+] T lymphocytes, including Th1 and Th17 [27,28]. Although the etiology still unclear, increasing evidence showed that genetic and environmental factors were associated with MS [29,30]. It is well known that the complex pathogenesis of MS mainly includes demyelination and inflammation [31]. Transplantation or remyelination could promote the recovery of neurological function of EAE [32,33]. However, the transplantation of stem cells is the focus on the clinical treatment of MS [34,35].

Increasing evidences have demonstrated that BMSCs promoted remyelinate axons and neurological function recovery in EAE animal model [36]. However, the mechanism of differentiation of BMSCs on EAE animal models remains to be studied. Myelination is carried out by oligodendrocytes in the central nervous system. Myelination is a modified expanded glial membrane that wraps around axons to achieve rapid saline-alkali nerve conduction and axon integrity [37]. Oligodendrocytes are derived from oligodendrocyte progenitor cells (OPC), and oligodendrocytes hold the capacity to proliferate, migrate and differentiate into myelinating oligodendrocytes. After demyelination, OPC is recruited to differentiate into myelinated oligodendrocytes, which then act on remyelination to protect axons from degeneration [38]. Therefore, oligodendrocytes are of great significance for remyelination. Astrocytes originate

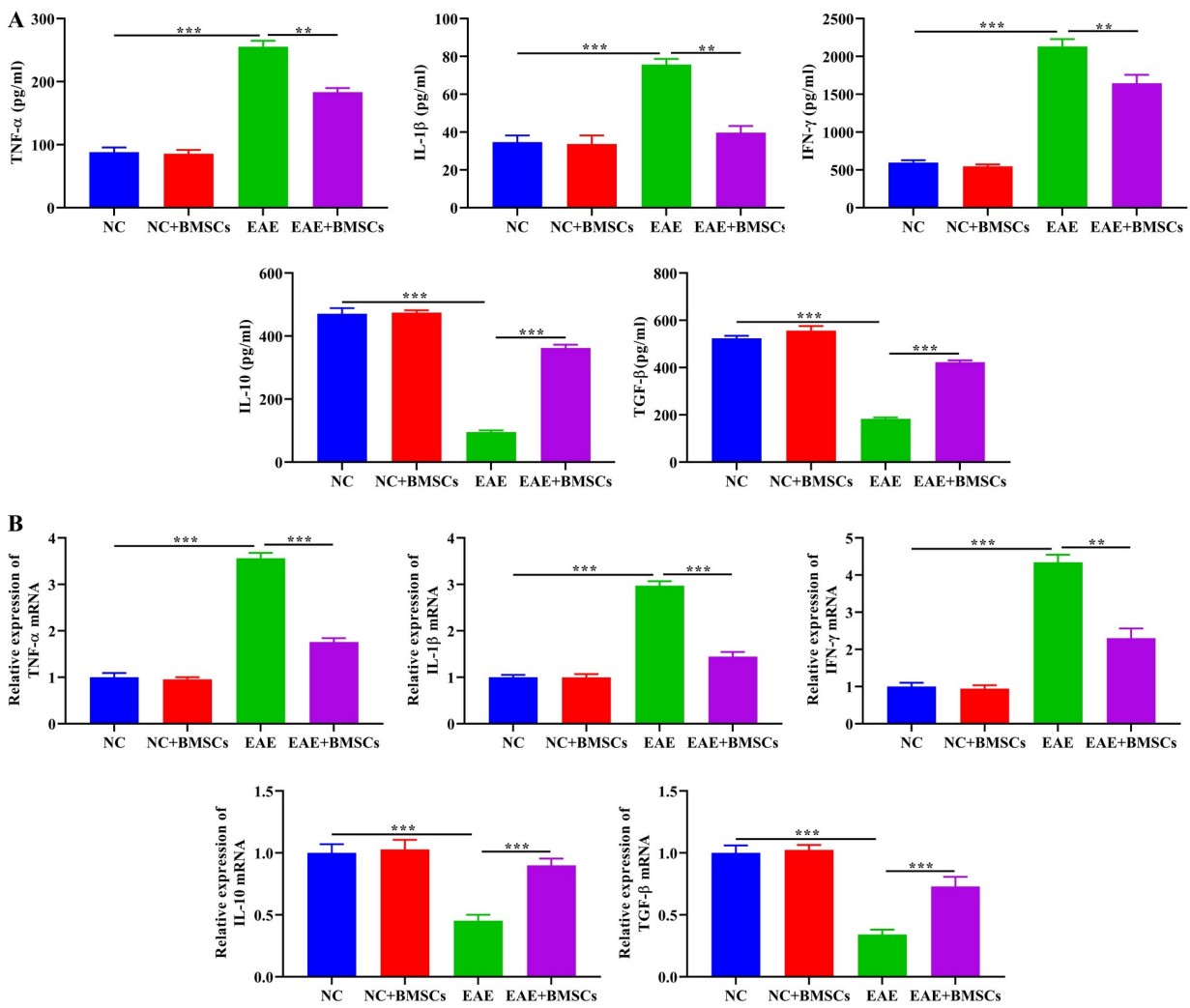

**Fig 5. Effect of BMSCs transplantation on inflammatory factor expression during EAE progression in mice.** On the 16th day post-immunization, (A) ELISA was performed to detected the levels of pro-inflammatory factors (TNF-α, IL-1β and IFN-γ) and anti-inflammatory factors (IL-10 and TGF-β) of mice serum of each group. (B) RT-qPCR indicated the expression of TNF-α, IL-1β, IFN-γ, IL-10 and TGF-β mRNA of mice spinal cord of each group. In all cases, Values are mean ± SD (n = 15 for each group; $^{**}P<0.01$, $^{***}P<0.001$).

from neural embryonic progenitor cells arranged in the embryonic neural tube cavity [39]. It is currently recognized academically that astrocytes can support the function of oligodendrocytes. As early as 1984, studies have shown that type 1 astrocytes could expand O-2A progenitor cells from the optic nerve of newborn rats [40]. Later, Bhat and Pfeiffer observed [41] that extracts from astrocytes-rich cultures stimulated the differentiation of oligodendrocytes, thus supporting the concept that astrocytes played an active role in myelination. In addition, astrogliosis is one of the pathological features of MS. Astrocytes regulate the integrity of the blood brain barrier (BBB) by regulating the transport of peripheral immune cells [42], and can secrete a large number of chemokines and cytokines with pleiotropic functions [43], which lays an active role in promoting demyelination. So, this study investigated whether the transplanted BMSCs can differentiate into neurons, astrocytes and oligodendrocytes to affect the inflammatory invasion and demyelination of EAE mice. However, intravenous injection of

BMSCs could not across the BBB, which is a key problem in the future. In the present study, BMSCs were injected into the lateral ventricle to detect the markers of oligodendrocytes, neurons and astrocytes. We found that BMSCs could be induced to differentiate into neurons, astrocytes and oligodendrocytes *in vitro* and *in vivo*. Furthermore, BMSCs transplantation alleviated clinical signs, infiltration of inflammatory cells and demyelination of EAE mice, and down-regulated the expression levels of pro-inflammatory factors TNF-α, IL-1β and IFN-γ, while increasing the expression level of anti-inflammatory factors IL-10 and TGF-β. Although stem cell transplantation is advanced, the success rate of stem cell transplantation is relatively low [44]. Some studies have evaluated the effect of stem cell transplantation on the clinical treatment of MS [45–47]. Thus, improving the success rate of stem cell transplantation could be used as a means of clinical treatment of MS.

Although in this study, we lacked the specific mechanism of neurons, astrocytes and oligodendrocytes differentiated by BMSCs for the treatment of MS. As discussed earlier, neurons, astrocytes and oligodendrocytes are beneficial for MS treatment, which indicates that at least part of the improvement of MS by transplanted BMSCs is due to differentiation into neurons, astrocytes and oligodendrocytes *in vivo*. As for the specific mechanism, it will be the focus of our future research.

## Conclusion

In conclusion, this study found that BMSCs could alleviate the inflammatory response and demyelination in EAE mice, which may be achieved by the differentiation of BMSCs into neurons, astrocytes and oligodendrocytes in EAE mice.

## Supporting information

**S1 Checklist. Plos one humane endpoints checklist.**
(DOCX)

**S1 File. Ethical approval.**
(ZIP)

## Acknowledgments

We thank members of our laboratory for providing technical advice and encouragement.

## Author Contributions

**Conceptualization:** Guo-yi Liu.

**Data curation:** Fan-yi Kong, Jia Geng.

**Formal analysis:** Fan-yi Kong.

**Investigation:** Shu Ma, Li-yan Fu.

**Methodology:** Shu Ma, Li-yan Fu.

**Project administration:** Li-yan Fu.

**Supervision:** Fan-yi Kong, Jia Geng.

**Validation:** Li-yan Fu.

**Visualization:** Yan Wu.

**Writing – original draft:** Guo-yi Liu.

**Writing – review & editing:** Yan Wu, Jia Geng.

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
