## [Decision Letter · Decision Letter 0]

15 Jan 2021

PONE-D-20-34923

BMSCs differentiated into neurons, astrocytes and oligodendrocytesalleviatedthe inflammation and demyelination of EAE mice models

PLOS ONE

Dear Dr. 

Thank you for submitting your manuscript to PLOS ONE. After careful consideration, we feel that it has merit but does not fully meet PLOS ONE’s publication criteria as it currently stands. Therefore, we invite you to submit a revised version of the manuscript that addresses the points raised during the review process.

We look forward to receiving your revised manuscript.

Kind regards,

Rosanna Di Paola, MD

Academic Editor

PLOS ONE

Journal Requirements:

3. Please ensure that you refer to Figure 5 in your text as, if accepted, production will need this reference to link the reader to the figure.

Reviewers' comments:

Reviewer's Responses to Questions

**Comments to the Author**

1. Is the manuscript technically sound, and do the data support the conclusions?

Reviewer #1: Partly

Reviewer #2: Partly

2. Has the statistical analysis been performed appropriately and rigorously? 

Reviewer #1: I Don't Know

Reviewer #2: Yes

3. Have the authors made all data underlying the findings in their manuscript fully available?

Reviewer #1: Yes

Reviewer #2: No

4. Is the manuscript presented in an intelligible fashion and written in standard English?

Reviewer #1: Yes

Reviewer #2: No

5. Review Comments to the Author

Reviewer #1: The authors suggest that MSCs injected into the lateral ventricle in mice immunized to elicit MOG peptide EAE become neurons, astrocytes, and oligodendrocytes, and suppress clinical and histological EAE. This would be interesting, but there are, unfortunately, a number of flaws in experimental design and data presentation that decrease enthusiasm for this manuscript,

a) There is no control group in which MSCs were injected into unimmunized mice,

b) The method by which the MSCs were labeled prior to transplantation is not detailed.

c( The numbers of mice in the EAE experiments are not specified.

d) A MOG peptide EAE model in which all untreated mice died by day 17 post-immunization is most unusual, and deserves explanation. It is possible that what the authors meant to show in Figure 4A is the % of mice that got sick, not the percent that survived, since it is clear from Figures 4B and 4C that many mice survived, and it is specified in the text that no mouse became sicker than grade 4.

e) The immunohistological data presented in Figure 3A-C are poorly presented and hence are difficult to interpret. Furthermore, where those sections were in the CNS is not specified.

f) The histological section shown for EAE in Figure 4E appear to have been damaged

Reviewer #2: BMSCs differentiated into neurons, astrocytes and oligodendrocytes alleviated the inflammation and demyelination of EAE mice models by Gen et. al. The authors tackle an important and unmet area of multiple sclerosis. In this study, stem cells differentiated into CNS cells (neurons, oligodendrocytes and astrocytes) were injected intracerebrally to show that 1) viable transplanted cells were retained within the CNS, 2) the procedure modulated the systemic inflammatory response, and 3) resulted in amelioration of the EAE clinical phenotype. The concept of the study is innovative and novel. However, there are several issues that need to be addressed before it can be considered for publication.

1) Overall the experimental details are missing, which makes it hard for the reviewer to evaluate the findings and interpretations. Although the authors cited previous publications on stem cell culture and differentiation, details on how the stem cells were harvested, number of seeding cells, differentiation and monitoring procedures, % differentiation, time of harvest were missing. Immunolabeling for cell-specific markers (GFAP, O4, etc) were done, however, the authors also failed to functionally characterize the differentiated cells. In addition, presence of transplanted cells were shown by IHC but the location, whether they were integrated and /or functional was unclear. In addition, the number of cells transplanted and when during the EAE disease course, the percentage of total injected cells retained within the CNS, the ratio of differentiated neurons, astrocytes and oligodendrocytes were not mentioned.

2) Blurry figures and figure legends lacking details. Photomicrographs also lack scale bars, arrow, etc.

3) Proper controls are lacking (non-EAE, BMSC injection).

4) Gender of EAE mice was not mentioned. However, BMSCs from male mice only were harvested and the rationale was not explained.

5) Pertussis toxin is usually injected twice at day 0 and day 2 in MOG35-55-immunized EAE model. Authors also included a third injection at day 7. Please explain the rationale.

6) EAE is an immune model where the immuno-pathogenesis initiates in the peripheral immune system. Please explain how CNS injection of differentiated stem cells modulate the systemic immune response as shown by down regulation of pro-inflammatory circulating cytokine.

6. PLOS authors have the option to publish the peer review history of their article (what does this mean?). If published, this will include your full peer review and any attached files.

Reviewer #1: No

Reviewer #2: No

---

## [Author Response · Author response to Decision Letter 0]

12 Mar 2021

Titled: BMSCs differentiated into neurons, astrocytes and oligodendrocytes alleviated the inflammation and demyelination of EAE mice models

Manuscript Number: PONE-D-20-34923

Journal: PLOS ONE

Dear Editors and Reviewers:

On behalf of my co-authors, we thank you very much for giving us an opportunity to revise our manuscript, we appreciate editor and reviewers very much for their positive and constructive comments and suggestions on our manuscript entitled “BMSCs differentiated into neurons, astrocytes and oligodendrocytes alleviated the inflammation and demyelination of EAE mice models”. Those comments are all valuable and helpful for revising and improving our paper, as well as the important guiding significance to our researches. We have studied comments carefully and have made correction which we hope meet with approval. Revised portion are marked in RED in the paper. The main corrections in the paper and the responds to the reviewer’s comments are as followings:

Journal Requirements:

RE: Thanks for your comment. We have revised it according to PLOS ONE's Style Requirements.

RE: Thank you for reminding me. We have deleted the Ethics Statement outside the Methods section.

3. Please ensure that you refer to Figure 5 in your text as, if accepted, production will need this reference to link the reader to the figure.

RE: We have added the corresponding position of Figure 5 to the manuscript Result section. Thanks again for the warm reminder.

Reviewers' comments:

Reviewer's Responses to Questions

Comments to the Author

1. Is the manuscript technically sound, and do the data support the conclusions?

Reviewer #1: Partly

Reviewer #2: Partly

2. Has the statistical analysis been performed appropriately and rigorously?

Reviewer #1: I Don't Know

Reviewer #2: Yes

3. Have the authors made all data underlying the findings in their manuscript fully available?

Reviewer #1: Yes

Reviewer #2: No

4. Is the manuscript presented in an intelligible fashion and written in standard English?

Reviewer #1: Yes

Reviewer #2: No

Review Comments to the Author

Reviewer #1: The authors suggest that MSCs injected into the lateral ventricle in mice immunized to elicit MOG peptide EAE become neurons, astrocytes, and oligodendrocytes, and suppress clinical and histological EAE. This would be interesting, but there are, unfortunately, a number of flaws in experimental design and data presentation that decrease enthusiasm for this manuscript,

a) There is no control group in which MSCs were injected into unimmunized mice,

RE: Thanks for your comments. In in vivo, we added experiments in the NC group (unimmunized mice) and NC+BMSCs group (BMSCs injected into unimmunized mice). Please see Fig.3, Fig.4 and Fig.5 for details. Moreover, the corresponding parts in the manuscript have also been revised.

b) The method by which the MSCs were labeled prior to transplantation is not detailed.

RE: Thanks, we have already supplemented these contents in the Materials and Methods, please see line 115-121.

c) The numbers of mice in the EAE experiments are not specified.

RE: We shown in the Materials and Methods (line 123-124) and figure legend (line 497-534) that we have 15 mice in each group. Thank you very much for your advice.

d) A MOG peptide EAE model in which all untreated mice died by day 17 post-immunization is most unusual, and deserves explanation. It is possible that what the authors meant to show in Figure 4A is the % of mice that got sick, not the percent that survived, since it is clear from Figures 4B and 4C that many mice survived, and it is specified in the text that no mouse became sicker than grade 4.

RE: We are very grateful for your correction, and Figure 4A is a major carelessness. As you said, Figure 4A shows a ‘Percentage disease-free’. We have corrected the figure and the corresponding text (line 265-270). Thanks again for your comment.

e) The immunohistological data presented in Figure 3A-C are poorly presented and hence are difficult to interpret. Furthermore, where those sections were in the CNS is not specified.

RE: Thank you very much for your valuable comments. Firstly, we reformatted Figure 1-5 with AI, which will make Figure more conducive to observation, and marked the location of the staining is the hippocampus and cortex. Secondly, due to the addition of NC and NC+BMSCs groups, we analyzed the differentiation of BMSCs into neurons, astrocytes and oligodendrocytes in the NC+BMSCs and EAE+BMSCs groups, and found that under EAE conditions, BMSCs are easier to differentiate into neurons, astrocytes and oligodendrocytes in vivo (Figure 4C). At the same time, we also analyzed the difference in fluorescence intensity of each cell marker in each group (Figure 4B). These results explain to a certain extent that the differentiation of BMSCs is of positive significance in EAE treatment.

f) The histological section shown for EAE in Figure 4E appear to have been damaged

RE: We have replaced Figure 4E. However, maybe we sliced and cut a bit thinly, or severe spinal cord injury in the EAE group. The tissue was somewhat incomplete in all our slices of EAE group. The replaced photomicrograph is the most complete one we can find. Thanks again for your comments during your busy schedule.

Reviewer #2: BMSCs differentiated into neurons, astrocytes and oligodendrocytes alleviated the inflammation and demyelination of EAE mice models by Gen et. al. The authors tackle an important and unmet area of multiple sclerosis. In this study, stem cells differentiated into CNS cells (neurons, oligodendrocytes and astrocytes) were injected intracerebrally to show that 1) viable transplanted cells were retained within the CNS, 2) the procedure modulated the systemic inflammatory response, and 3) resulted in amelioration of the EAE clinical phenotype. The concept of the study is innovative and novel. However, there are several issues that need to be addressed before it can be considered for publication.

1) Overall the experimental details are missing, which makes it hard for the reviewer to evaluate the findings and interpretations. Although the authors cited previous publications on stem cell culture and differentiation, details on how the stem cells were harvested, number of seeding cells, differentiation and monitoring procedures, % differentiation, time of harvest were missing. Immunolabeling for cell-specific markers (GFAP, O4, etc) were done, however, the authors also failed to functionally characterize the differentiated cells. In addition, presence of transplanted cells were shown by IHC but the location, whether they were integrated and /or functional was unclear. In addition, the number of cells transplanted and when during the EAE disease course, the percentage of total injected cells retained within the CNS, the ratio of differentiated neurons, astrocytes and oligodendrocytes were not mentioned.

RE: We have supplemented the specific steps of BMSCs differentiation in the method section, please see line 135-148.

Since funds have been exhausted, we cannot perform functional characterization of differentiated cells. At present, we can only achieve that BMSCs can differentiate into neurons, astrocytes and oligodendrocytes in vitro. 

Regarding ‘presence of transplanted cells were shown by IHC but the location, whether they were integrated and /or functional was unclear’, this is also a question we have been thinking about. As we all know, neurons, astrocytes and oligodendrocytes have positive significance for remyelination and nerve damage repair in MS treatment. Therefore, we proved that BMSCs can differentiate into these three kinds of cells in vivo, which shows that the transplantation of BMSCs can play a positive role in the treatment of MS in this way. Although we have not studied the specific mechanism, this is what we will explore later.

Regarding ‘the number of cells transplanted and when during the EAE disease course, the percentage of total injected cells retained within the CNS, the ratio of differentiated neurons, astrocytes and oligodendrocytes were not mentioned’, we have been trying to solve this problem before. However, due to financial and technological constraints, we can only study to this point. We can only know how many cells we have transplanted, but we do not have the technology to assess how many cells in the CNS are retained or differentiated. In addition, due to the addition of NC and NC+BMSCs groups, we analyzed the differentiation of BMSCs into neurons, astrocytes and oligodendrocytes in the NC+BMSCs and EAE+BMSCs groups, and found that under EAE conditions, BMSCs are easier to differentiate into neurons, astrocytes and oligodendrocytes in vivo (Figure 4C). At the same time, we also analyzed the difference in fluorescence intensity of each cell marker in each group (Figure 4B). These results explain to a certain extent that the differentiation of BMSCs is of positive significance in EAE treatment.

In short, the main purpose of this article is to explain that the MS therapeutic effect of BMSCs transplantation is achieved at least in part by differentiation into neurons, astrocytes and oligodendrocytes. As for the specific mechanism, it will be the focus of our future research.

Thank you for pointing out the flaws in our article, which is also a question we have been thinking about. We have done our best to make the revision. Thanks again.

2) Blurry figures and figure legends lacking details. Photomicrographs also lack scale bars, arrow, etc.

RE: Thank you very much for your valuable comments. Firstly, we reformatted Figure 1-5 with AI, which will make Figure more conducive to observation. Moreover, we have enriched the content in Figure legend to show more details. There is scale bar in Photomicrographs, but they are a bit small. Secondly, we used arrows to mark the main observation positions of Photomicrographs.

3) Proper controls are lacking (non-EAE, BMSC injection).

RE: Thanks for your comments. In in vivo, we added experiments in the NC group (unimmunized mice) and NC+BMSCs group (BMSCs injected into unimmunized mice). Please see Fig.3, Fig.4 and Fig.5 for details. Moreover, the corresponding parts in the manuscript have also been revised.

4) Gender of EAE mice was not mentioned. However, BMSCs from male mice only were harvested and the rationale was not explained.

RE: We added the gender of the mice in line 92, that all our experiments use male mice. Moreover, we have supplemented the specific process of BMSCs isolation in the method section (line 106-121). Thank you for your valuable suggestions.

5) Pertussis toxin is usually injected twice at day 0 and day 2 in MOG35-55-immunized EAE model. Authors also included a third injection at day 7. Please explain the rationale.

RE: Thanks for your correction. This due to the ambiguous description of our method, and we have already revised it, please see line 123-133.

6) EAE is an immune model where the immuno-pathogenesis initiates in the peripheral immune system. Please explain how CNS injection of differentiated stem cells modulate the systemic immune response as shown by down regulation of pro-inflammatory circulating cytokine.

RE: Thanks again for your comments during your busy schedule. As we said before, we do not have enough funds to support this article to conduct deeper mechanism research. We admit that the current article does have many flaws, but this is also something that point the way for our future research.

---

## [Editor Report · Decision Letter 1]

27 Apr 2021

BMSCs differentiated into neurons, astrocytes and oligodendrocytes alleviated the inflammation and demyelination of EAE mice models

PONE-D-20-34923R1

Dear Dr. 

We’re pleased to inform you that your manuscript has been judged scientifically suitable for publication and will be formally accepted for publication once it meets all outstanding technical requirements.

Kind regards,

Rosanna Di Paola, MD

Academic Editor

PLOS ONE
---

## [Editor Report · Acceptance letter]

30 Apr 2021

PONE-D-20-34923R1 

BMSCs differentiated into neurons, astrocytes and oligodendrocytes alleviated the inflammation and demyelination of EAE mice models 

Dear Dr. Geng:

I'm pleased to inform you that your manuscript has been deemed suitable for publication in PLOS ONE. Congratulations! Your manuscript is now with our production department. 

Kind regards, 

on behalf of

Dr. Rosanna Di Paola 

Academic Editor

PLOS ONE